# Platinum and Palladium Complexes as Promising Sources for Antitumor Treatments

**DOI:** 10.3390/ijms22158271

**Published:** 2021-07-31

**Authors:** Robert Czarnomysy, Dominika Radomska, Olga Klaudia Szewczyk, Piotr Roszczenko, Krzysztof Bielawski

**Affiliations:** Department of Synthesis and Technology of Drugs, Medical University of Bialystok, Kilinskiego 1, 15-089 Bialystok, Poland; dominika.radomska@umb.edu.pl (D.R.); szewczyk.o.k@gmail.com (O.K.S.); roszczenko.piotr@gmail.com (P.R.); krzysztof.bielawski@umb.edu.pl (K.B.)

**Keywords:** platinum complexes, palladium complexes, anticancer treatments

## Abstract

There is a need for new, safer, and more effective agents to treat cancer. Cytostatics that have transition metals at their core have attracted renewed interest from scientists. Researchers are attempting to use chemotherapeutics, such as cisplatin, in combination therapy (i.e., in order to enhance their effectiveness). Moreover, studies are being carried out to modify molecules, by developing them into multinuclear structures, linking different compounds to commonly used drugs, or encapsulating them in nanoparticles to improve pharmacokinetic parameters, and increase the selectivity of these drugs. Therefore, we attempted to organize recent drug findings that contain palladium and platinum atoms in their structures.

## 1. Introduction

Platinum-based compounds have been widely used in cancer chemotherapy. The primary compound that led to the development of this group is cisplatin. It was first synthesized in 1844 by an Italian chemist, Michele Peyrone, which is why it was originally called Peyrone’s chloride [1]. The antitumor potential of cisplatin was found by accident and reported by Rosenberg in 1965. Rosenberg used platinum electrodes when studying the effects of low-voltage alternating currents on the growth of *Escherichia coli* bacteria. Inhibition of cell proliferation without a corresponding inhibition of bacterial growth was observed due to the reaction of platinum from the electrodes with NH_4_Cl, resulting in the formation of cisplatin, which was the cause of these changes [2]. Rosenberg and his partners began experimenting on mice with leukemia and sarcoma [3], which led to the start of clinical trials in 1971. This resulted in the official introduction of cisplatin for the treatment of testicular and ovarian cancer in 1978 [4], a milestone in chemotherapy. Nowadays, cisplatin is used in various kinds of tumor diseases. Clinical guides mainly talk about treating testicular [5], ovarian [6], bladder [7], head and neck [8], lung [9], and cervical cancer [10]. Cisplatin, despite its undeniable benefits, has a dark side, i.e., its adverse effects. For drugs commonly used in platinum-based chemotherapy, many specific side effects may occur in the treated patient, such as cardiotoxicity, nephrotoxicity, ototoxicity, hematological toxicity, hepatotoxicity, gastrointestinal toxicity, and neurotoxicity. Unfortunately, tumor cells also develop resistance; thus, we are seeing a decline in efficacy and an associated reduction in patient survival, even though there is a good initial response to treatment [11].

Transition metals are a promising base for anticancer drugs. Complexes containing palladium in their structures are closely related to their platinum analogs, because chemical and physical properties of platinum and palladium are comparable to each other. Palladium, same as platinum, is contained in anticancer structures in metallic or ionic form (Pd^2+^ or Pd^4+^). The metal itself can also be used in radiotherapy, such as ^103^Pd. Similar chemical and physical properties of platinum and palladium suggest that they can be used interchangeably in analogous compounds of antitumor properties [12]. Both metals belong to the same group in the periodic table—platinum-group elements (PGE) and the bonds formed by them have similar lengths. For example, the M-Cl bond length (M = Pt or Pd) in the compound K_2_[MCl_4_] is 2.316 Å and 2.318 Å for platinum and palladium, respectively [13]. Despite the undeniable similarity, the kinetics of the palladium compounds are 10^5^ times faster [12]. This means that they are much more reactive, necessitating the stabilization of the palladium ion by using specific chelating ligands. The differences in the reactivity of these two transition metals should be found in the field of quantum chemistry. Palladium has the electron configuration [Kr]4d^10^, while platinum [Xe]4f^14^5d^9^6s^1^ impacts the properties of these metals. Platinum that possesses a 5d orbital has higher ionization potential compared to palladium that possesses a 4d orbital (the 6s orbital has lower energy), because this shell is located further from the positively charged nucleus, making it easier to “detach” an electron, and the bonds formed in platinum compounds are stronger. In addition, the greater stability of platinum compounds is related to the slight difference in the energy of the 5d and 6s orbitals, because the electrons of the 5d orbital can more easily form bonds [14].

Toxicological tests have shown 10 times lower toxicity of palladium compounds than platinum-based compounds in studies performed on rats [15]. The reduced toxicity of palladium complexes to normal tissues may be explained by the impossibility of the sulfhydryl groups to substitute for the tightly bound chelate ligands of Pd(II) when the compound interacts with proteins inside the cells [16].

Today, platinum-based cytostatics are much more common, but they have low solubility in water, significant toxicity, and studies have documented the development of cancer cell resistance. The problematic nature of platinum-based drugs has stimulated the search for alternatives, which may be palladium compounds. Because of the complexity of the issue, we have attempted to organize the existing data.

## 2. Cisplatin and Its Derivatives Commonly Used in Cancer Treatment

Despite the undeniable advantages of cisplatin, toxicity and increased cell resistance are significant issues in effective chemotherapy. Therefore, there is a search for new drugs as modifications of cisplatin. In order to explain what factors affect the properties of a molecule, first, we have to understand its mechanism of action. A complex that has metal at its center not only induces desirable processes, such as apoptosis, but also leads to other cell death pathways [17]. A model of a perfect compound based on a transition metal would exclude alternative pathways because of the disturbance of homeostasis, of an organism that would be subjected to chemotherapy, and the associated complications of treatment. Nowadays, special emphasis is placed on the substitution of one (or both) of the N-donor ligands, by chelating oxalate, carboxylate, or glycolate [18]. It has led to the development of cisplatin derivatives, which are shown in Figure 1. Among them, we distinguish oxaliplatin and picoplatin (Figure 1, structures 5 and 6), which represent the next generation of platinum drugs.

Carboplatin (Figure 1, structure 2) represents the second generation of platinum-based drugs. It was designed as a prodrug [19]. Compared to cisplatin, it was modified by substitution of the chloride group by the 1,1-cyclobutanedicarboxylate ligand, to obtain a more stable compound, whose conversion to reactive compounds was prolonged. The modifications resulted in lower toxicity than the original. The mode of action of carboplatin is analogous to that of cisplatin because it binds to DNA. Unfortunately, the effect of cross-resistance of tumor cells was noticed [20].

In the 1980s, analogs of carboplatin were developed that contained a cyclobutane ring in their structures. In this way, zaniplatin, miboplatin, and enloplatin (Figure 1, structures 7 and 8) were created. Nedaplatin and lobaplatin (Figure 1, structures 3 and 4) are structure-linked compounds of the second generation of platinum compounds. Importantly, lobaplatin was indicated as effective in cancers showing resistance to cisplatin [21].

Oxaliplatin (Figure 1, structure 5) is the third generation of platinum compounds. In contrast to previous derivatives, it was not found to cause resistance in cancer cells. The molecule was modified by combining cisplatin with the large ligand diaminocyclohexane and oxalate. With this modification, a lack of nephrotoxicity and myelosuppression has been reported, but neurotoxicity is significant. Oxaliplatin induces oxidative stress in tumor cells to a greater extent than it leads to cell death by blocking DNA replication [22].

## 3. Modifications of Platinum- and Palladium-Based Molecules

There is considerable interest from scientists in transition metal based anticancer drugs. This interest is based on numerous studies surrounding the synergism of existing drugs with potentially supporting substances, as well as the structural modification of molecules. In this section, we review novel platinum- and palladium-based molecules and their modifications that exhibit anticancer activity, which have been reported on in recent years (i.e., primarily in 2020 and 2021).

### 3.1. Platinum-Based Compounds

Modern antitumor drugs (Figure 2) include substances that exhibit (and have a strong affinity for) carbonic anhydrase inhibition. The targeting is caused by an acidic environment and hypoxia in the tumor cells. Due to this, these substances suppress tumor metabolic pathways, which results in increased therapeutic effects proven on MDA-MB-231 breast cancer cells in comparison to cisplatin and oxaliplatin. With the modification, significantly higher selectivity and cytotoxicity of the new drugs on cancer cells under hypoxia are observed [23].

New complexes of Pt(IV) with dihydro-2-quinolone, namely DHQLO (Figure 3, structures 11 and 12), are also being investigated. In the study, mono DHQLO complexes were found to exhibit higher effectiveness than binary complexes. The alkyl chain connecting the platinum core and DHQLO was also shown to have a significant effect on the results. The use of a butyryl linker causes a similar effect as cisplatin or oxaliplatin, but demonstrates better selectivity, and is active against cells resistant to cisplatin. In addition to binding to the DNA of tumor cells, DHQLO derivatives display mitochondria-damaging activity and, thus, can activate tumor cell apoptosis [24].

Investigations into new drugs are directed towards specific ways of binding DNA, mainly basing themselves on giving the molecule alkylating abilities. This is achieved by selecting appropriate ligands for platinum complexes, such as those based on terpyridine with mustard substituents, or derivatives obtained by hydroxylation (Figure 4, structures 13 and 14). The designed compounds had potent antiproliferative effects on four cancer cell lines—human colorectal (HCT116), non-small cell lung (NCI-H460), cervical (SiHa), and colon (SW480) cancer, with compound **13** exhibiting higher efficacy. Acceptable stability of the complex was proven, which provides hope for the discovery of another way to transition metal-based drugs [25].

Efforts are being made to decrease the negative effects of known antitumor agents. One example of a modification aimed at establishing this state is via the synthesis of new platinum complexes, with glycine derivatives as ligands, with the pattern [Pt(R-amine)_2_(R-gly)]NO_3_ (Figure 5). The R-amine position contains propylamine, tert-Pentylamine, or isopentylamine, and the R-gly position contains propylglycine, tert-pentylglycine, or isopentylglycine. The MTT assay was used to evaluate cytotoxicity on the MCF-7 human breast cancer cell line. In this investigation, cisplatin was used as a reference compound. Based on the nucleophilicity predictions of the new complexes and the LADME (liberation, absorption, distribution, metabolism, and excretion) results, the anticancer properties of the new complexes should be equal or superior to the reference. The propyl derivative exhibited the best activity. The research revealed the interaction between the new complex and DNA. Circular dichroism (CD) spectrum confirmed the bonding of the positively charged complex through electrostatic interactions, which are less effective than hydrogen interactions, from which we can conclude that the side effects of glycine derivatives can be significantly reduced compared to cisplatin [26].

In recent years, there has been a significant increase in combining benzimidazole and platinum. This combination is interesting because benzimidazole and its derivatives exhibit a variety of beneficial effects, such as anti-inflammatory or antiproliferative activity. In this type of complex, the biological activity is affected by the presence of carbon located between two nitrogen atoms. As this combination is quite common, it was modified, and its effect on brain cells was testes, since knowledge on this issue is quite limited. Therefore, glioblastoma, a highly aggressive cancer with a poor prognosis, and neuroblastoma, the most frequent solid tumor in pediatric patients, were targeted. For these malignancies, therapy is mainly based on cisplatin, which offers an opportunity for the development of new, more effective, and safer medications. In this study, three new platinum complexes with 2,6-di-tert-butyl-4-(1-phenyl-1*H*-benzimidazol-2-2yl) phenol (Figure 6, structure 15), N,N-dimethyl-4-(1-phenyl-1*H*-benzimidazol-2-yl) aniline (Figure 6, structure 16), and 4-(1*H*-benzimidazol-2-yl)-N,N-dimethylaniline (Figure 6, structure 17) were used for their cytotoxicity against brain tumor (U87 and SHSY-5Y) cell lines. Two of them (complex **15** and **17**) were proven effective on these lines at significantly lower concentrations than other previously known synthesized compounds described in the literature. In particular, complex **15** exhibited significantly higher selectivity for the U87 and SHSY-5Y lines in comparison to the reference (cisplatin). Disubstituted benzimidazole platinum derivatives have potential as a therapy against glioma and neuroblastoma cells [27].

There are also other reports in the field of brain tumor research. In a study conducted by Rimoldi et al. (2018) [28], a novel synthesized cationic platinum(II) complex (caPt(II)-complex, Figure 7) exhibited more potent cytotoxic activity against U87 MG glioblastoma cells than the reference compound (cisplatin). The IC_50_ of caPt(II)-complex was 19.85 ± 0.97 µM whereas that of cisplatin was 54.14 ± 3.19 µM. Moreover, despite the similar antiangiogenic properties of both compounds, the novel cationic platinum(II) complex was found to be more stable compared to cisplatin.

Triple-negative breast cancer is a type of tumor that does not respond to hormones—estrogen and progesterone—and does not depend on the human epidermal growth factor receptor 2 (HER2) protein. It is not possible to treat this type of cancer with hormonal therapy or drugs that target the HER2 receptor. This cancer has molecular features that allow for an easy epithelial-to-mesenchymal transition (EMT) and, therefore, increased invasiveness and a chance of metastasis. Chinese research scientists have managed to create a platinum(IV) conjugate consisting of ketoprofen and cisplatin called ketoplatin. Via this combination, they obtained a prodrug with less systemic toxicity and an increased anticancer effect (about 50-fold) compared to cisplatin. Additionally, by affecting the inflammatory environment through modulation of cyclooxygenase-2 (COX-2) signaling, the progression of EMT was suppressed [29].

A new platinum complex with folic acid (FA) (cis-[Pt(NH_3_)_2_FA], Figure 8) was developed. Using MTT assay on MCF-7 breast cancer cells, it was observed that this complex had improved cytotoxicity and induced apoptosis more potently compared to cisplatin, additionally enhancing its cellular uptake. Molecular docking confirmed that the platinum complex may form a very stable complex with folate receptors [30].

Platinum compounds, in addition to the mononuclear forms reviewed earlier, can be decidedly more complicated. The formulation of these molecules is intended to bind more efficiently to the DNA of tumor cells and may provide a far more selective anticancer activity. It is worth noting that platinum complexes are cationic in nature, and their properties are due to the structure of these compounds. Analyzing the structure of most platinum-based complexes, it is observed that the metal is bonded to two donor nitrogen atoms. This results in the formation of bidentate chelating ligands that have stable coordination complexes with metals, including transition metals, such as platinum, which gives these compounds new biological properties. Apart from the action of these complexes on cell DNA and other molecular targets, as mentioned earlier, these compounds are characterized by their selectivity against cancer cells. This is related to the transporters that are involved in the transport of these complexes into the cell—cationic platinum complexes are transported via organic cation transporters (OCTs) that are overexpressed in cancer cells, for example, colorectal cancer [31].

Gęgotek et al. [32] decided to create berenil-platinum(II) complexes (Figure 9) and investigate the apoptosis-related protein expression rates. The binuclear berenil-platinum(II) complexes more effectively mediate cellular oxidative modification and proapoptotic metabolism compared to cisplatin, especially in MCF-7 breast cancer cells. Pt_2_(isopropylamine)_4_berenil_2_ (Figure 9, structure 18) and Pt_2_(piperidine)_4_(berenil)_2_ (Figure 9, structure 19) complexes significantly impacted the cellular metabolism of estrogen-positive breast cancer cells. Thus, it may be beneficial to further research these compounds in regard to the discovery of effective anticancer drugs.

The binuclear platinum compounds can not only be targeted for anticancer activity, but may also inhibit neoangiogenesis. Novel binuclear complexes were synthesized with various pyridine-like binding ligands: 4,4′-bipyridine (Figure 10, structure 20), 1,2-bis(4-pyridyl)ethane, and (Figure 10, structure 21) 1,2-bis(4-pyridyl)ethene (Figure 10, structure 22). They interact with the phosphate backbone to form DNA-Pt adducts. This pairing provides a type of bonding not yet reported before, called fine-groove coating. They have also overcome cisplatin resistance and have higher in vivo antitumor efficacy in mice. Additionally, higher anti-angiogenic efficacy was proven when compared to the drug (sunitinib malate). A large therapeutic window was demonstrated, which is highly beneficial for cancer treatment. The study revealed that novel binuclear Pt(II) complexes may be new, effective, and safe anticancer agents [33].

A new monofunctional trinuclear platinum complex (MTPC) (Figure 11), can generate MTPC-DNA adducts via bifunctional and trifunctional cross-links. Hydrogen bonding analysis revealed that coordination of MTPC to DNA leads to the reduced thermal stability of base pairs and structural deformations of DNA. More significant DNA structural distortions were found to occur in trifunctional cross-linking than in bifunctional cross-linking. All MTPC-DNA adducts exhibit distinct conformational variations, including bending and twisting motions. The results provide a deeper understanding of MTPC-induced DNA structure deformation with various cross-links at the nucleotide scale, which provides a basis for the development of novel anticancer compounds [34].

### 3.2. Palladium-Based Compounds

Palladium compounds can be divided into two groups. The first has one palladium atom in its core. This type of compound also contains ligands with proven biological activity [35]. These ligands can be pyridine, quinoline, or their analogs. The second group of palladium derivatives with anticancer properties are molecules containing two palladium atoms in the core. As in the previous group, ligands are selected based on their potentially beneficial properties. Cytostatics, containing transition metals in their core, interact with the DNA of cancer cells. This occurs by producing both covalent and non-covalent bonds. In contrast to cisplatin, palladium derivatives bind to the oligonucleotide [d(CGCGAATTCGCG)]_2_, which results in blocking the replication of DNA. Both groups of derivatives show high selectivity to other oligonucleotide end regions. Due to bypassing the resistance of cancer cells, it has been found that they can also enter into non-covalent, electrostatic, and hydrogen bonds with DNA. It is advantageous to introduce a monoethylphosphine or diethylphosphine group, which allows for increased solubility of the resulting complexes [36].

Besides intercalation into DNA, other targets of the molecular mechanism of palladium(II) complex activities have been described. One of them is the induction of apoptosis proceeding through both extrinsic (death receptors-mediated) and intrinsic (mitochondrial) pathways. It was observed that these compounds caused upregulation of Bax protein and downregulation of Bcl-2 protein, which resulted in decreased mitochondrial potential and a release of cytochrome c, activating the caspase cascade, proving that apoptosis occurs via the mitochondrial pathway. Meanwhile, in the death receptor-mediated pathway, it was reported that Pd(II) complexes caused an increase in the expression of cell death receptor genes DR4 and DR5. Apart from defects in mitochondria, existing damage of the endoplasmic reticulum (ER) was also reported and was caused by oxidative stress. Excessive generation of reactive oxygen species (ROS) was the result of Pd(II) reacting with thiol groups of proteins, including antioxidant system proteins (glutathione-S-transferase (GST), glutathione peroxidases (GPxs), catalase (CAT), and glutathione (GSH)), and a decrease in their levels in the cell. In addition, cell arrest in the G_2_/M phase of the cell cycle was also noted [37,38]. Despite the many similarities between palladium(II)-based complexes and their platinum analogs, both in terms of chemistry and mechanism of action, there are some minor differences between them. First, as mentioned earlier, their complexes have different stabilities [38]. Secondary, their mechanisms of molecular action are slightly different. Platinum analogs, the same as Pd(II) complexes, also intercalate into DNA, cause oxidative stress, and induce apoptosis via the extrinsic and intrinsic pathways. The difference between them is likely only in the phase of the cell cycle in which the cells are arrested. There are reports that, besides arresting cells in the G_2_/M phase, platinum analogs can also arrest them in the G_1_ and S phase of the cell cycle [39].

Al-Saif et al. [40] synthesized six new PdCl_2_-based palladium complexes—one with picolinic acid (Figure 12, structure 23), isonicotinic acid (Figure 12, structure 24), nicotinamide (Figure 12, structure 25), and these complexes with an additional caffeine ligand (Figure 12, structures 26–28). In cytotoxicity assays on colorectal adenocarcinoma (Caco-2) and breast cancer (MCF-7) cell lines, each of the synthesized compounds was found to have this activity. However, among the six complexes, the most potent activity was demonstrated by the complex with picolinic acid (Figure 12, structure 23), which indicates that it may be a potential anticancer compound used against human colorectal adenocarcinoma and breast cancer [40].

One of the palladium(II) complexing ligands selected by the researchers was 2-hydrazinopyridine conjugated with malonate, oxalate, and pyrophosphate ligands (Figure 13, structures 29–30). The structural analysis indicated that the complexes were tetragonal and exhibited diamagnetic properties. Simulation results additionally demonstrated that the compounds were expected to be stable. The activity of the complexes was tested on breast (MCF-7), hepatocellular (HepG-2), prostate (PC-3), and larynx (HEP-2) cancer cell lines. The oxalate ligand compound that was created showed the best antiproliferative and cytotoxic activity against all cell lines screened. Importantly, all tested compounds had greater activity in comparison to the reference compound (vinblastine sulfate). Additional studies are needed to uncover the dependence of antitumor activities on the structure of the complexes [41].

Another interesting type of structure is Pd(II) complexes with 2-acetyl-5-methylthiophene and thiosemicarbazones (Figure 14, structures 32–34). Thiosemicarbazones coordinate the metal molecule via nitrogen and sulfur. They have been confirmed to have inhibitory activity on five human carcinoma cell lines (colon, cervix, hepatocellular, breast, and prostate). It was noted that the existence of a coordinated chloride ion could affect the antiproliferative properties of the derivatives. It was also observed that the derivative, which is toxic to both cancer and non-cancer cells, has only one sulfur atom in its structure, while the derivative that has selective effects on cancer cells has two sulfur atoms. Curiously, these complexes are not electrolytes, so that the chloride ions are strongly coordinated and the thiosemicarbazones are trapped by thiol tautomer anions [42].

The combination of palladium(II) with ferrocene Schiff base named Fc-Ppd-2 resulted in satisfying cytotoxicity against renal (ACHN) and breast (MCF-7) cancer. This complex has been found to put the cell into a state of physiological cell death, which is apoptosis. By this process, we eliminate the undesired consequences of inflammation in the tumor area. Studies have confirmed more than 25-fold higher effectiveness compared to cisplatin. Moreover, this compound is effective against cisplatin-resistant cancer cells [43].

The existing problem with lung cancer therapy, or for that matter, the lack of an appropriate line of treatment for small-cell lung cancer, has led researchers to search for new medications. Two bis(phosphinite) palladium(II) complexes were synthesized and investigated against non-small-cell lung cancer (NSCLC) and small-cell lung cancer (SCLC). Surprisingly, both complexes showed activity against NSCLC (H1975 and HCC78) and SCLC (H209, N417) cell lines. These complexes introduced tumor cells into the apoptosis pathway, which is especially preferable for treatment. In particular, this cell death occurs through stimulation of p21 and Bax protein expression. Complex **35** (Figure 15) was found to have the strongest cytotoxic and pro-apoptotic activity on both tumor types, which is registered for the first time in history and, more significantly, offers a chance for successful treatment of SCLC. It was also shown that despite its inferior efficacy, complex **36** (Figure 15) was the most effective of all in inducing apoptosis in lung cancer cells. What is remarkable is that, despite previous studies, ruthenium derivatives formed analogically acted weaker, just like the reference compound, i.e., cisplatin. This proves that palladium derivatives can be more effective than ruthenium ones when additional side groups are suitably matched. Created complexes provide safer, more effective treatments for patients [44].

During Saeb Aliwaini’s research [45], compound AJ-5 (Figure 16) was synthesized as a promising cytostatic agent for the treatment of advanced melanoma and breast cancer. It is a compound that has two palladium atoms in its structure. AJ-5 was found to initiate DNA double-strand breaks and induction of the p38 protein, leading to cell cycle arrest and, consequently, initiation of apoptosis. The compound bound to cancer cell DNA and caused inhibition of cellular metabolism [45].

The synthesis of binuclear analogs of the formulation [{Pd(en)Cl}_2_(μ-L)](NO_3_)_2_ (where L is quinoxaline, quinazoline, and phthalazine) (Figure 17, structures 37–39) was attempted. The interaction of the complexes with calf thymus DNA was investigated. The antiproliferative and apoptotic activities of Pd(II) compounds were investigated on lung and colorectal cancer cell lines. Researchers found decreased efficacy in colorectal cancer cell lines and higher efficacy in lung cancer cells of these analogs compared to cisplatin. All complexes induced apoptosis, however, the quinazoline-containing compound exhibited the best response [46].

Spermine is an organic chemical compound from the polyamine group. Binuclear complexes of Pd(II) with amines, such as spermine (Figure 18), are being considered as promising candidates for clinical trials. In studies, they demonstrate beneficial levels of cytotoxicity against cancer cells exhibiting resistance. The dinuclear chelate, also called Pd_2_Spm, penetrates cells by passive diffusion. The pharmacokinetic profile and distribution of Pd_2_Spm is advantageous for cancer treatment. This compound induces a much faster metabolic response compared to cisplatin, so that the performance of the tested organs returns to its original state much earlier; thus, patients are expected to recover sooner than with cisplatin treatment [47].

A novel dinuclear palladium(II) complex with nitroimidazole as a ligand (Figure 19) incubated with MDA-MB-231 human breast cancer cells induces a three-fold increase in caspase 8, 9, and 10 activities in tumor cells compared to cisplatin. The presented results indicate that the novel palladium(II) complex with nitroimidazole has high proapoptotic activity because caspases 8, 9, and 10 belong to the initiators of apoptosis. The increase in the amounts of their active forms indicate that, in MDA-MB-231 breast cancer cells, the process of physiological cell death has been launched. It is supposed that the active form of this complex is formed by the cleavage of the triazine bond in the berenil molecule that leads to the release of the transition metal (Pd) [48]. Whereas in the next stage, the released Pd is likely transported into the cell, the same as the two basic transporters as cisplatin—human copper transporter protein 1 (Ctr1) and organic cation transporter 2 (OCT2) [49,50]. The investigated compound is a promising alternative to currently used cytostatic drugs, due to almost three-fold stronger stimulation of the apoptosis process and bypassing the resistance of tumor cells [51].

## 4. Synergism of Biologically Active Compounds with Platinum- and Palladium-Based Molecules

Anticancer treatment has changed rapidly throughout the decades. This is related to the persistent work by scientists in overcoming the resistance of neoplastic cells to chemotherapeutic agents. One important aspect includes the reduction of registered side effects and increasing the safety of the therapy. One way to improve treatment parameters is to combine one substance with another with frequently proven biological activity. The natural candidates for combined treatments are agents that exhibit cytotoxic activity. Many of these combinations are being tested in clinical trials, in order to confirm or deny the efficacy of these treatments [52].

Paclitaxel is an effective antineoplastic drug, but it has a low therapeutic index, so it has the same burdensome side effects as cisplatin. Combination therapy with these two drugs was developed, leading to clinical trials of this pairing [53].

Tegafur–uracil (UFT) composed of tegafur and uracil is an oral chemotherapeutic drug. With its low toxicity comes low effectiveness in malignancies. Studies suggest that it is beneficial to combine UFT with cisplatin in patients [54].

Doxorubicin is an effective, but unfortunately significantly toxic, cytostatic. It can be used successfully on its own, but when combined with cisplatin, it was demonstrated to significantly extend patients’ lives. Combination therapy with these cytostatic agents has intermediate strength, but side effects are well tolerated by patients [55].

Gemcitabine has both antiviral and antitumor activity. It does not exhibit significant toxicity. Previous reports show that the combination of cisplatin with this nucleoside analog results in increased survival with moderate toxicity [56].

Vitamin D has been known for many years, but reports suggest that vitamin D may affect the inhibition of cell proliferation and induction of apoptosis in tumor cells. The combination of vitamin D and cisplatin showed synergistic regulation of the NF-κB pathway, modulated by lipocalin-2 (LCN2) in oral squamous cell carcinoma [57].

Plant-derived substances, such as lycopene, are another group of compounds that act synergistically with cytostatics, offsetting side effects. Lycopene is a natural carotenoid that is found in fruits and vegetables. Previous studies have demonstrated its anticancer and anti-inflammatory effects, although its mechanism is not fully recognized. Its synergistic effect with cisplatin against HeLa cervical cancer cells has been proven. It was found to sensitize the tumor to cisplatin by decreasing cell viability; regulation of apoptotic proteins also decrease inflammatory response and oxidative stress [58].

Melatonin is a natural hormone produced by the human body. In addition to its known properties, it was revealed that it can also have neuroprotective, nephroprotective, or hepatoprotective effects. This is achieved by the reduction of oxidative stress, which is caused by destructive factors, such as oncological drugs. Stimulation of Nrf2, which results in decreasing levels of ROS and malondialdehyde (MDA), was observed [59]. This allows mitigating the side effects of chemotherapy.

Cryptotanshinone is a quinoid diterpenoid isolated from the roots of the plant *Salvia miltiorrhiza* (family *Lamiaceae*). It is a compound with proven multidirectional activities, including anti-inflammatory, antioxidant, and anticancer properties against breast, lung, and liver cancer [60]. Cryptotanshinone was also proven to exhibit anticancer effects by inducing apoptosis, modulating cell proliferation, and inhibiting ovarian cell metastasis. Additionally, it may increase the sensitivity of ovarian cancer cells to cisplatin treatment in vitro [61]. Blocking the Nrf2 signaling pathway is responsible for these effects [62]. This signaling pathway is also affected by resveratrol [63], berberine [64], and diallyl trisulfide [65].

The overexpression of cytochrome P450 CYP1B1 was identified as one of the factors responsible for cell resistance to cisplatin. *Glycyrrhiza glabra* (family *Fabaceae*) and quercetin isolated from it was shown to have powerful inhibitory potency on CYB1B1 and CYP1A1 enzymes; therefore, overcoming cisplatin resistance [66]. Glycyrrhizin and lamivudine in combination contribute to the reduction of cisplatin efflux from hepatocellular carcinoma cells; thus, cell resistance to cisplatin treatment is reduced by modulation of multidrug resistance-related proteins (MRPs) [67].

Luteolin and apigenin are compounds classified as flavonoids. They have strong effects on Oct-4/Sox2/c-Myc expression. Studies have shown that they cause decreased cell proliferation, arrest cells in the G_2_/M and S phase of the cell cycle, and enhance apoptosis. Luteolin and apigenin both activated the expression of Oct-4, Sox2, and c-Myc in a time- and dosage-dependent manner. In conclusion, this study revealed that luteolin and apigenin successfully activated Oct-4/Sox2 signaling, suggesting that these flavonoids may act synergistically in combination with cytostatic agents [68].

Polyphenols have a multidirectional activity that we can use in relieving the side effects of chemotherapy and in lowering the resistance of cytostatic drugs. Researchers are investigating the combinations of compounds, such as flavonols with metals in the core. Available results indicate that this combination improves biological properties, such as enhancing anti-inflammatory effects of quercetin with zinc, versus using the compound by itself. By developing such a hybrid, we notice an increase in the solubility of polyphenols, which leads to enhanced bioavailability of these compounds. This provides a unique opportunity to exploit the anticancer potential of synergistically active metal and plant-derived substances [69].

Palladium complexes show significant efficacy against prostate and lung cancer cells. Transition metal combinations have been used as biologically active molecules, demonstrating their unique ability to bind various biological targets. One important issue in the design of new cytostatic molecules is coordinating ligands around the palladium center [36].

Mononuclear palladium complexes bonded to aromatic ligands, demonstrating hydrophilic properties and strong coordination effects. Some binuclear palladium complexes have interesting spherical structures and good antitumor activities. An attempt to modify natural drugs with Pd^2+^ led research to a new pathway. A palladium derivative created with the ligand 2,6-dimethyl-4-nitro-pyridine (dmnp) (Figure 20) showed significantly greater activity than cisplatin in three of four breast cancer cell lines tested [70].

In light of the increasing evidence reported on the anticancer properties of palladium complexes, a series of novel palladium(II) compounds were synthesized on the square planar. All compounds showed significant activity against human cervical (HeLa) and breast (MDA-MB-231) cancer cell lines, while they had a minor effect on normal fibroblasts (MRC-5). Moreover, all complexes exhibited the ability to direct cells into the apoptotic pathway, arresting the cell cycle in G_0_/G_1_ and S phase, as well as the G_2_/M phase. The study was performed in relation to cisplatin and, additionally, the combination of these compounds was tested. Importantly, the combination of the tested complexes with cisplatin at low concentrations proved effective, with moderate to strong synergistic effects. Thus, it appears that it will be possible to use these complexes in combination therapy with cisplatin in the future, which would lead to a reduction of its used doses. In this way, it could result in a limitation of the side effects of this well-known anticancer drug [71].

Another approach combining platinum and palladium is the synthesis of nanoparticles, containing both of these metals. Ghosh et al. [72] obtained platinum, palladium, and platinum–palladium nanoparticles (PtNps, PdNps, and Pt-PdNps) by a green synthesis method, using *Dioscorea bulbifera* (family *Dioscoreaceae*) tuber extract. In the analysis carried out using transmission electron microscopy (TEM), it was observed that the obtained Pt-PdNps were irregular shape with a diameter of 20–25 nm. In addition, it was found that Pt-PdNps exhibited the strongest cytotoxic effect among all tested nanoparticles against HeLa cervical cancer cells (25.75% of viable cells at a concentration of 10 µg/mL). Another research team used *Peganum harmala* (family *Nitrariaceae*) seed alkaloids for the green synthesis of the same nanoparticles as Ghosh, obtaining spherical Pt-PdNps with a smooth margin, and a size in the range of 28.1–38.9 nm, which were also evaluated by the TEM technique. To investigate the anticancer properties of these three nanostructures, Fahmy et al. used lung (A549) and breast (MCF-7) cancer cell lines. In the performed MTT assay, it was again found that Pt-PdNps exhibited the highest cytotoxicity among three obtained nanoparticles (IC_50_ was 8.8 µm/mL and 3.6 µg/mL for lung and breast cancer, respectively) [73]. To summarize these two studies, it can be concluded that the combination of platinum and palladium increases their anticancer activity, and this effect is synergistic.

## 5. Nanocarriers of Platinum and Palladium Complexes

In recent years, we have observed “technological leaps” in regard to delivering therapeutic agents into the body. With drug carriers, we are observing much lower toxicity and higher specificity of drug activity. This often involves encapsulating the drug particle in a cage that carries it to the appropriate site of activity [74].

Nanoparticles (NPs) have been developed to overcome the limitations of free molecules and to defeat biological barriers. As lipid, polymer, and inorganic NPs are created with increasing precision, they can begin to be optimized for improved drug delivery. Due to the degradation of some polymers at low pH, they are being used in chemotherapy. This is connected with the acidic environment of tumor cells, which is caused by lactic acidosis. Nanotechnology can help defeat large-scale problems, such as biodistribution and lower-scale barriers, such as targeting specific cells or molecular transport to organelles. NPs have the potential to increase the stability and solubility of drugs, promote transport across the membranes, and enhance circulation time to improve safety and efficacy. Many earlier NPs were incapable of overcoming these biological barriers to deliver drugs, but more recent NP projects have progressed in controlled synthesis strategies. These NPs can therefore be used as more complex systems, as carriers to alter several pathways, maximize therapeutic effects against specific molecules, target particular stages of the cell cycle, or defeat drug resistance mechanisms [75]. Another property of NPs is exploited—some of them can generate heat in the presence of an alternating external magnetic field. This method involves the administration of NPs that then selectively accumulate in tumor cells. The nanostructures generate local heat when an alternating magnetic field is applied, raising the temperature in the tumor environment. This results in increased immunogenicity of the cancer cells and leads to the body’s immune response against them. Ultimately, this process results in the selective death of cancer cells without affecting normal tissues [76]. Furthermore, the development of platinum-based photoactive prodrugs may also increase the selectivity and reduce the toxicity of tumor treatments available for laser devices [77].

Liposomes may be a specific cage for the drug. Cancer tumors have an abnormal vascular system that is characterized by increased permeability to molecules circulating in the blood, including liposomes, which leads to selective accumulation of these nanostructures inside neoplastic cells. Moreover, they can also protect the drug from deactivation by plasma proteins, and modifying their surface with various polymers provides the possibility of extending the circulation of the drug in the human body up to several months [78].

Liposomes are highly appealing for drug delivery. Unlike other types of NPs, the surface of a liposome is fluid, which allows for the dynamical organizing of targeting ligands for the optimal binding to receptors on the cell membrane. Lipids are an elemental ingredient of the cell membrane, and many of them are highly biocompatible. In addition, multiple types of pharmaceutical molecules, water-soluble and insoluble, can be encapsulated because they consist of a hydrophobic and hydrophilic region. Lipids can also be wrapped around inorganic and polymeric NPs to form supporting monolayers or bilayers of lipids. Compared to inorganic drug carriers, liposomes are softer and can be formed more easily Therefore, basic biophysical investigations can be useful in the rational formulation of drug design. A common lipid molecule has a hydrophilic main group and two hydrophobic tails. The charge of lipids and their chemical properties can be modulated by changing the main group, while the hydrophobic tails mainly regulate packing in membranes. The tail structures can be switched, giving different phase transition temperatures. These nanostructures appear ideal for the incorporation of platinum drugs. However, they may require the addition of biodegradable, biocompatible, low-immunogenicity polymers to enhance the control of release rate, biodistribution, and accumulation in tumors of the administered drugs. Furthermore, the stability of the micelles can be extended by cross-linking between the platinum and polymer chains [79,80].

Aroplatin is a platinum analog that is enclosed in a liposome. In vivo results suggest that it reaches efficacy in prolonging the life of mice with leukemia compared to free cisplatin. However in clinical trials, less than 6% of patients exhibited a response to this formulation and, additionally, there have been reports suggesting limited stability of the combination [81].

Lipoplatin is a formulation composed by cholesterol, 1,2-dipalmitoylsn-glycero-3-phospho-(1′-rac-glycerol) sodium salt (DPPG), hydrogenated soybean phosphatidylcholine (HSPC), and N-(carbonyl-methoxypolyethylene glycol 2000)-1,2-distearoyl-s*n*-glycero-3-phosphoethanolamine sodium salt (MPEG 2000-DSPE). The amount of cisplatin incorporated in the liposome forming phospholipids is 1:10. Due to the liposomes, the solubility of cisplatin increases significantly. Moreover, due to the structure, we get targeted therapy as it penetrates the imperfect vessels produced by tumors, and the addition of polyethylene glycols on the surface may increase the time that cisplatin stays in the human bloodstream. Preclinical studies of lipoplatin revealed less nephrotoxicity than cisplatin [82,83].

Nanoplatin, which is formulated with polymeric micelles, can enter the nucleus and cytoplasm of A549 non-small cell lung cancer cell line exhibiting cisplatin resistance. Ultimately, nanoplatin was found to exhibit a proapoptotic effect on non-small cell lung cancer cells. Relative to cisplatin, it significantly suppresses the proliferation of A549 lung cancer cells. This is probably due to stimulation of p53 protein and initiation of apoptosis [84].

Palladium nanostructures are promising photothermal factors, much more efficient than silver or gold nanoparticles. In the study, Pd-based nanosheets were used. This transition metal shows high absorption in the relevant spectral range, which means that it can effectively transfer absorbed energy as thermal energy to cancer cells. Additionally, the Pd nanosheets obtained in the study were analyzed by TEM, which revealed that these nanostructures had an average thickness of 1.8 nm [85].

Dendrimers are NPs with controlled mass, size, and functionality. They are branch-shaped polymers that structurally assume the form of a sphere. Symmetrically structured dendrimers can have various functional groups, depending on the generation of dendrons, which are branches created from polymer chains. They can provide an extremely interesting way to transport pharmaceuticals in the body, because not only can the drug be placed in the empty spaces between the branches, but with the functional groups, we can anchor the drug via covalent bonds. Properties of dendrimers vary according to what they are made of; however, the molecule itself is homogeneous. The most popular are PAMAM, which is composed of poly(amidoamine) or PPI of poly(propylene imine). Dendrimers protect the drug from inactivation by the body; they can release drug molecules in a controlled manner, and most significantly, they allow us to release medication at a specific location [86,87].

Due to the cancer cells resistance, targeted therapy is a preferable choice of treatment. This allows us to decrease the dosage of two drugs, which leads to a reduction in side effects. Unfortunately, difficulties are often encountered when drugs have different solubility or antagonistic nature. Because of these problems, it was decided to create polyamidoamine (PAMAM) dendrimers modified by hyaluronic acid (HA). These particles had four generations and were ended by amines. In turn, the attachment of HA by electrostatic interaction to the PAMAM dendrimer was aimed at both actively targeting this NP to cancer cells (via the CD44 receptor, the primary receptor for HA that is highly expressed in breast, lung, or pancreatic cancer cells), and extending its systemic retention time. Thus created NPs were filled with cisplatin (Pt) and doxorubicin (Dox). TEM analysis showed that all nanostructures were spherical. The PAMAM-Pt-Dox had a diameter of 91 ± 2.18 nm, and the HA functionalization (HA@PAMAM-Pt-Dox) increased its size to 106 ± 3.26 nm. Cytotoxicity assay showed that HA@PAMAM-Pt-Dox had significantly higher anticancer activity against human breast cancer cell lines MCF-7 and MDA-MB-231 at low concentrations. Due to dendrimer incorporation, significantly lower doxorubicin toxicity was reported. Moreover, the accumulation of HA@PAMAM-Pt-Dox particles in the tumor area was noted in the BALB/c nude assay. With these findings, we can hope to improve the efficacy of combination treatment with doxorubicin and cisplatin, with reduced adverse effects of these cytostatics [88].

An extremely innovative aspect of nanocarriers is the synthesis of jasmine-like hybrid Cu@L-Asp/PdPtNPs. Platinum and palladium in the molecule serve as signal enhancers. In scanning electron microscopy (SEM) analysis, the prepared Cu@L-Asps were characterized by a layered, jasmine-like structure of 1–2 µm size, and after Pd-Pt attachment, these Pd-PtNPs were uniformly placed on the surface of the hybrid Cu@L-Asp. The main target for the synthesis of the nanoparticle was the requirement for an efficient method to mark the amount of the intracellular SPOP protein. This protein, exhibiting poor expression levels, can cause increased proliferation and migration of ovarian cancer cells, which is the reason for metastasis creation. Therefore, quantitative detection of the SPOP protein in cells is important for cytology and for predicting the progression of ovarian cancer patients. Different concentrations of SPOP are detected by an antibody bound to the surface of Cu@L-Asp/Pd-Pt NPs, and the current signal decreases as the concentration of the labeled protein increases. This NP shows acceptable stability, its formation is reconstitutable, and its selectivity towards SPOPs is excellent as no signal interference by other intracellular particles was registered. Furthermore, it is possible to modify the NP and detect other proteins, which can be important factors in cancer diagnosis [89].

## 6. Platinum and Palladium Compounds in Clinical Trials

Cisplatin is used in the treatment of many types of cancer, but during monotherapy, with this drug, increasing resistance of tumor cells is observed. Mechanisms responsible for this phenomenon include decreased uptake and/or increased efflux of the drug, enhanced processes responsible for its metabolism/detoxification or DNA repair, and amplification of anti-apoptotic cellular mechanisms. Increasing the doses of platinum derivatives does not overcome the cancer cell resistance to this drug, but only intensifies the toxic effects against normal cells, which is associated with further side effects. For this reason, combination therapy is used to reduce the doses of platinum derivatives (reduction of side effects) and to overcome multidrug resistance. Currently, cisplatin is used in combination therapy with drugs, such as paclitaxel, 5-fluorouracil, or doxorubicin [90,91]. However, many investigators are looking for drugs that would act more potently in combination with platinum derivatives than those commonly used. Table 1 shows clinical trials from 2019 to 2021 that aim to test the efficacy of combining platinum-based chemotherapy with new compounds, including antibody-based targeted drugs.

Palladium is mainly used in radiotherapy [119]. It is used in the treatment of prostate cancer [120] and choroidal metastases [121], among others. There are also reports that palladium nanosheets can be safe and effective radiosensitizers [119]. Clinical trials that involve radiotherapy with ^103^Pd alone, or as a combination therapy, are presented in Table 2.

## 7. Summary

Transition metal-based drugs provide a promising basis for the development of new anticancer drugs. Continuous work by researchers across the world provides hope for more accurate and secure chemotherapy. We can increase the survivability of cancer patients via structural modifications, synergism with other drugs, or the use of carriers. Improving the quality of chemotherapy, by eliminating or reducing side effects, is an important goal that researchers should take on in the future. In vitro and in vivo studies, including clinical trials, with platinum- and palladium-based compounds, shed new light on these compounds. These derivatives appear to be very promising candidates for anticancer agents, but more extensive research in this area is required.

## Figures and Tables

**Figure 1 ijms-22-08271-f001:**
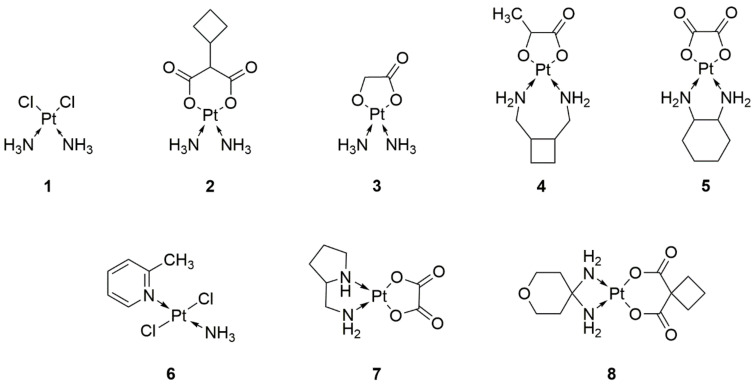
Structures of clinically used platinum anticancer drugs (**1**—cisplatin, **2**—carboplatin, **3**—nedaplatin, **4**—lobaplatin, **5**—oxaliplatin, **6**—picoplatin, **7**—miboplatin, **8**—enloplatin).

**Figure 2 ijms-22-08271-f002:**
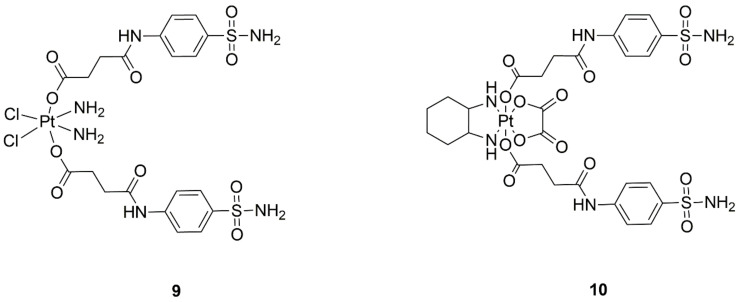
Chemical structures of carbonic anhydrase IX targeted platinum(IV) complexes with (**9**) cisplatin and (**10**) oxaliplatin core.

**Figure 3 ijms-22-08271-f003:**
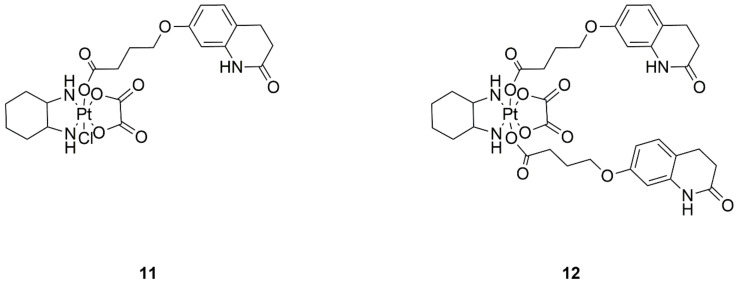
Structures of the new complexes of platinum(IV) with dihydro-2-quinolone—(**11**) mono DHQLO oxaliplatin and (**12**) dual DHQLO oxaliplatin complex.

**Figure 4 ijms-22-08271-f004:**
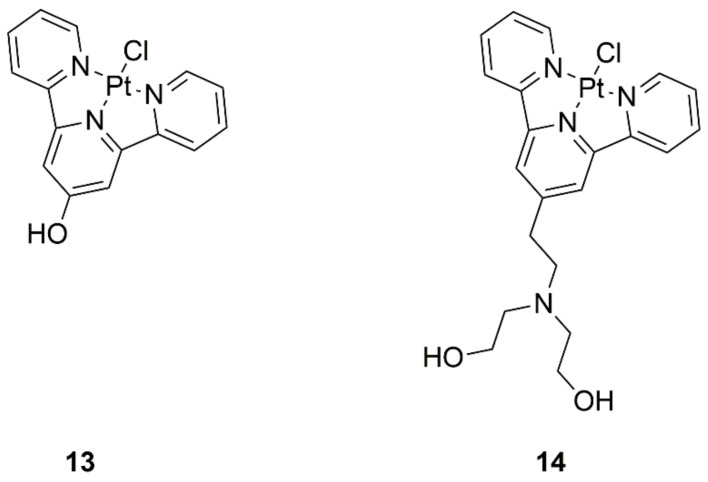
Structures of platinum(II) complexes with terpyridine derivatives exhibiting anticancer activity—(**13**) complex with 2,6-bis(2-pyridyl)-4[1*H*]-pyridone and (**14**) 2,2′-((2-([2,2′;6′,2″-terpyridin]-4′-yloxy)ethyl)azanediyl)bis-(ethan-1-ol).

**Figure 5 ijms-22-08271-f005:**
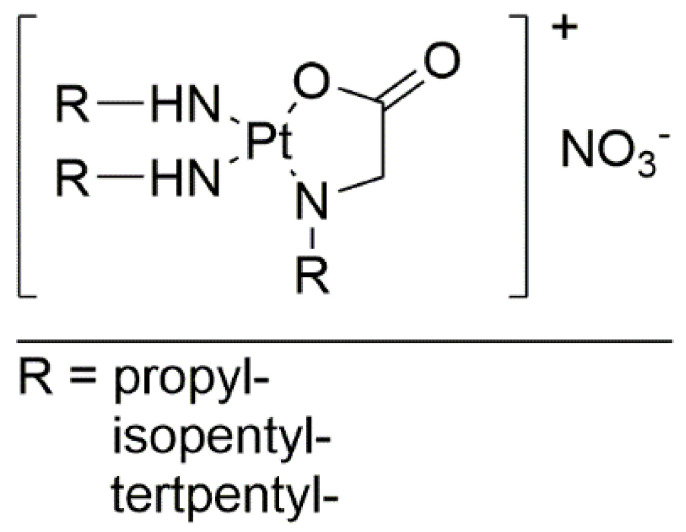
Novel platinum complexes with glycine derivatives.

**Figure 6 ijms-22-08271-f006:**
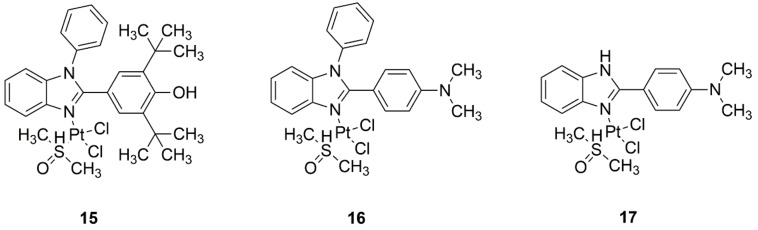
Structures of platinum(II) complexes with (**15**) 2,6-di-tert-butyl-4-(1-phenyl-1*H*-benzimidazol-2-2yl) phenol, (**16**) N,N-dimethyl-4-(1-phenyl-1*H*-benzimidazol-2-yl) aniline, and (**17**) 4-(1*H*-benzimidazol-2-yl)-N,N-dimethylaniline.

**Figure 7 ijms-22-08271-f007:**
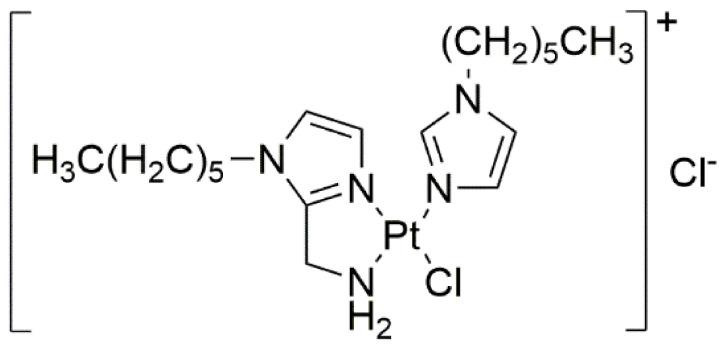
Novel cationic platinum(II)-complex (caPt(II)-complex).

**Figure 8 ijms-22-08271-f008:**
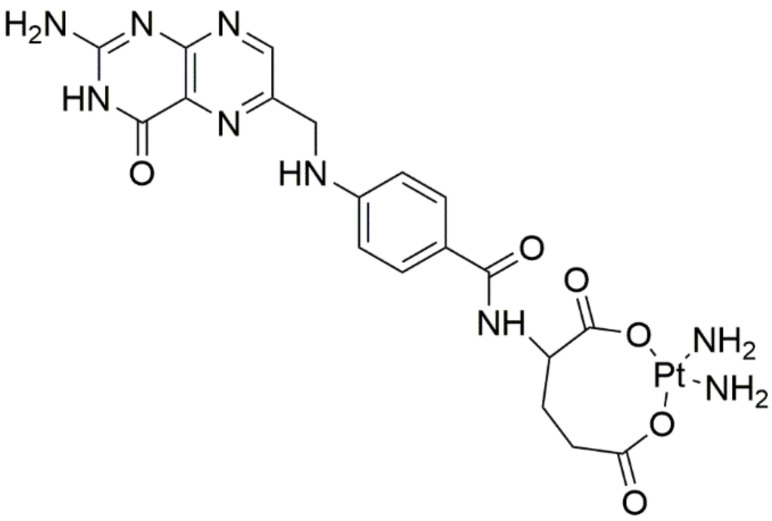
Novel platinum complex with folic acid.

**Figure 9 ijms-22-08271-f009:**
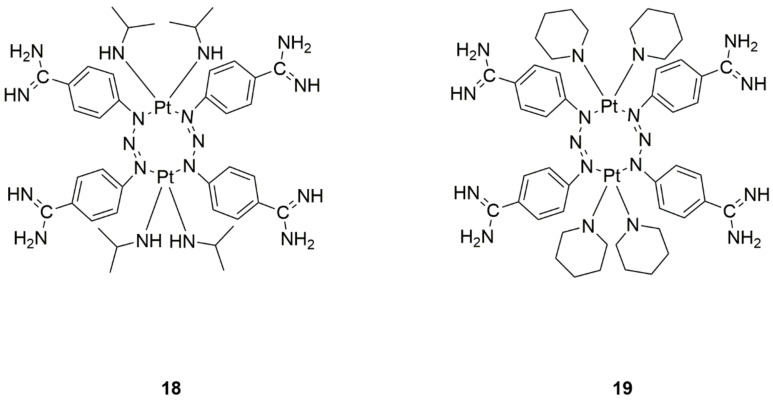
Structures of berenil-platinum(II) complexes (**18**—Pt_2_(isopropylamine)_4_berenil_2_ and **19**—Pt_2_(piperidine)_4_(berenil)_2_).

**Figure 10 ijms-22-08271-f010:**
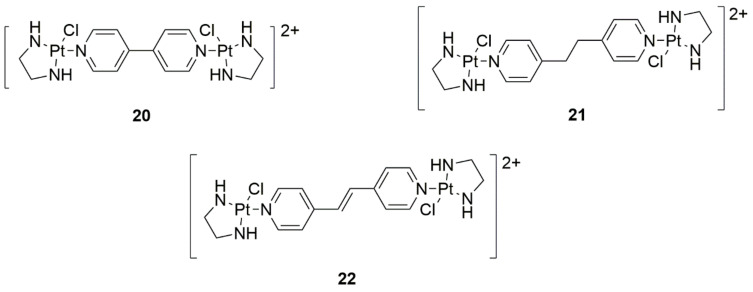
Structures of the new synthesized platinum structures with (**20**) 4,4′-bipyridine, (**21**) 1,2-bis(4-pyridyl)ethane and (**22**) 1,2-bis(4-pyridyl)ethene.

**Figure 11 ijms-22-08271-f011:**
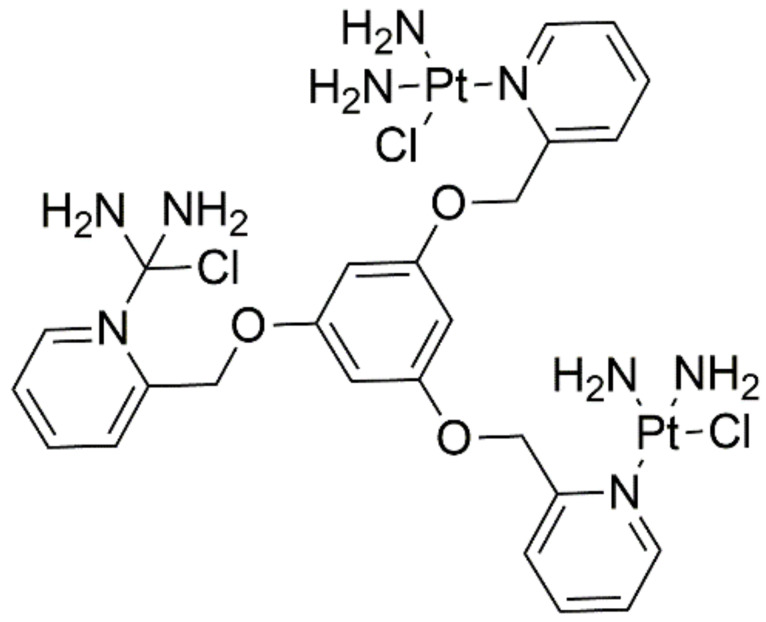
Structure of the new trinuclear platinum complex (MTPC).

**Figure 12 ijms-22-08271-f012:**
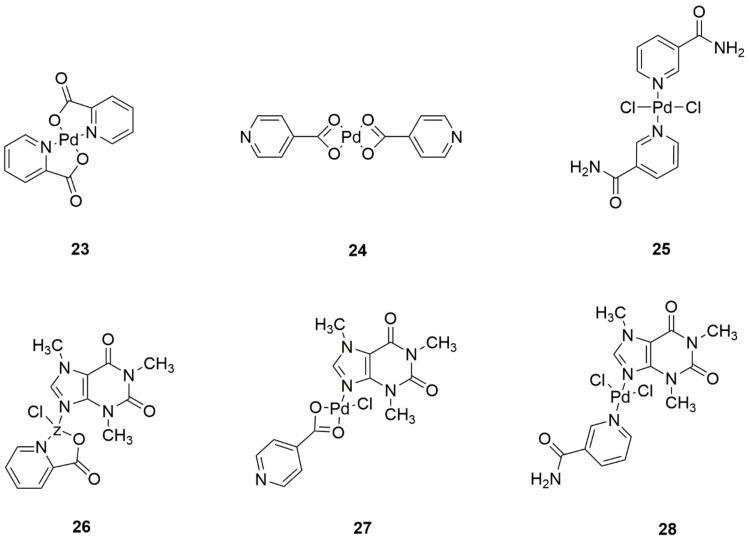
New palladium(II) complexes with (**23**) picolinic acid, (**24**) isonicotinic acid, (**25**) nicotinamide, and these complexes with an additional caffeine ligand (**26**–**28**).

**Figure 13 ijms-22-08271-f013:**
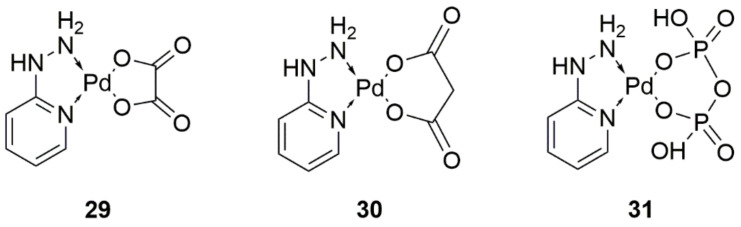
Structures of palladium(II) complexes with 2-hydrazinopyridine conjugated with (**29**) oxalate, (**30**) malonate, and (**31**) pyrophosphate ligand.

**Figure 14 ijms-22-08271-f014:**
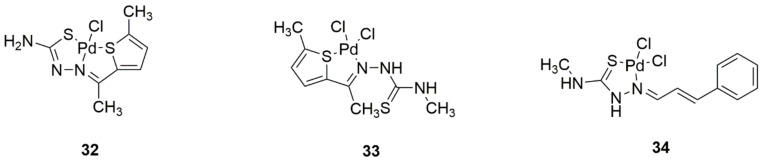
Structures of palladium (II) complexes with (**32**) 2-acetyl-5-methylthiopene thiosemicarbazone, (**33**) (2E)-2-[1-(5-Methyl-2-thienyl)ethylidene]hydrazinecarbothioamide, and (**34**) (2E)-2-[(2E)-3-phenyl-2-propen-1-ylidene]hydrazinecarbothioamide ligands.

**Figure 15 ijms-22-08271-f015:**
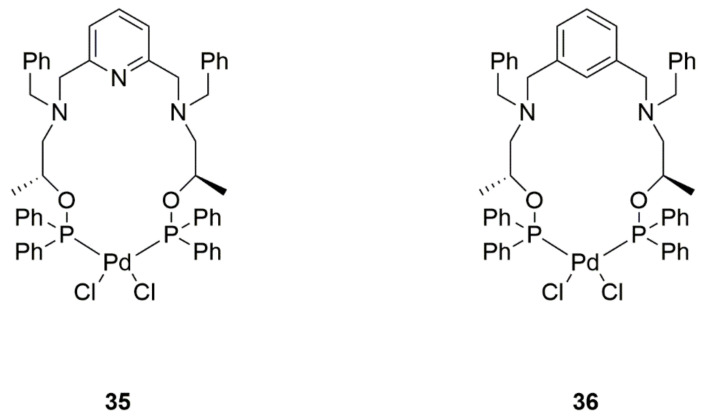
Structures of bis(phosphinite) palladium(II) complexes.

**Figure 16 ijms-22-08271-f016:**
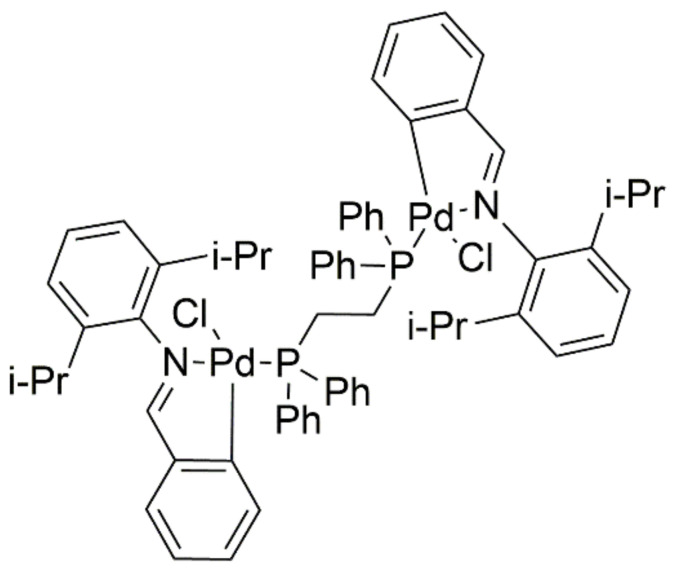
Complex AJ-5—binuclear palladacycle complex.

**Figure 17 ijms-22-08271-f017:**
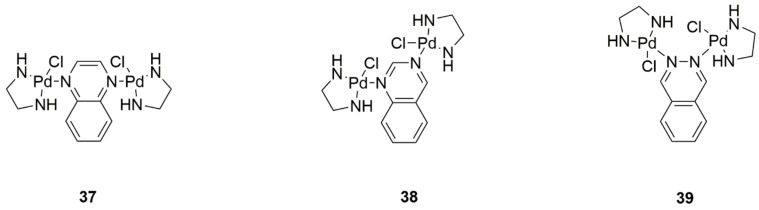
Palladium(II) binuclear complexes with quinoxaline (**37**), quinazoline (**38**), and phthalazine (**39**).

**Figure 18 ijms-22-08271-f018:**
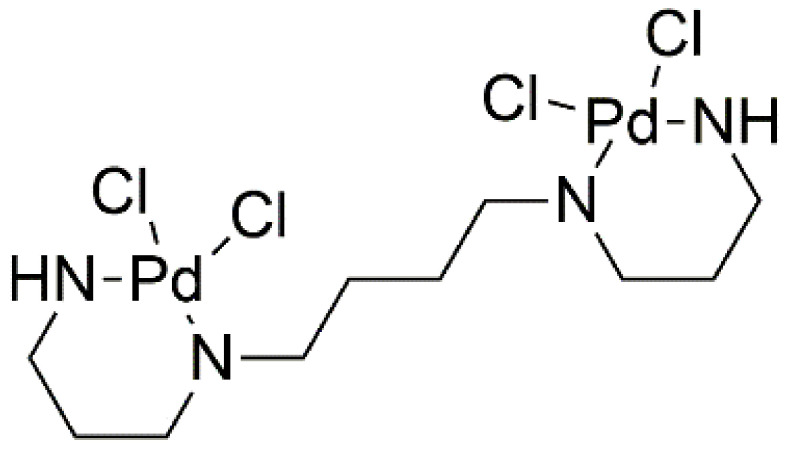
Structure of dinuclear palladium (II)—spermine complex (Pd_2_Spm).

**Figure 19 ijms-22-08271-f019:**
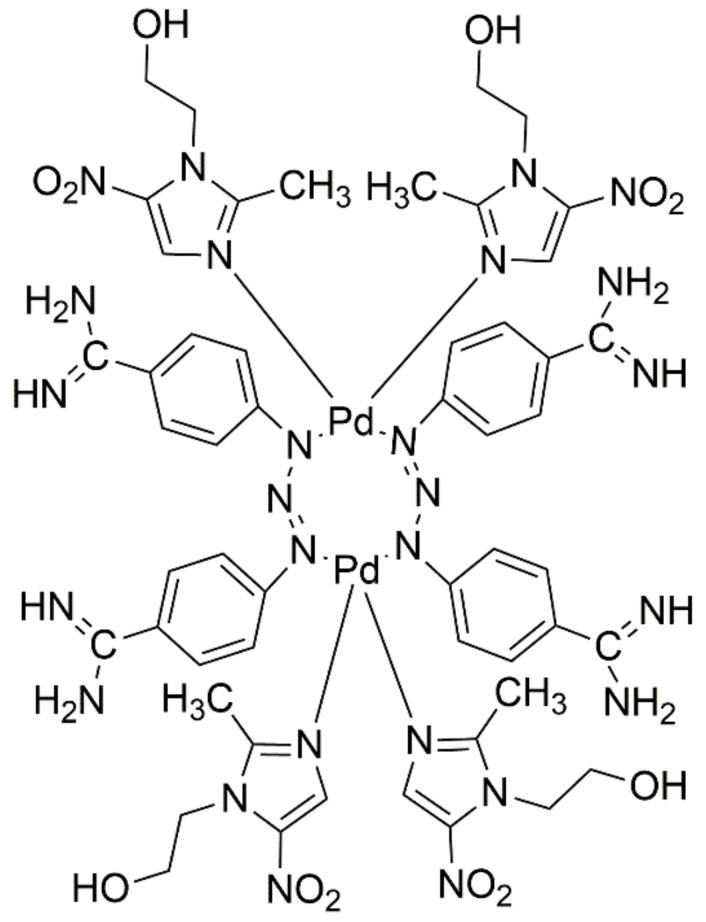
Structure of dinuclear novel palladium (II) complex with nitroimidazole.

**Figure 20 ijms-22-08271-f020:**
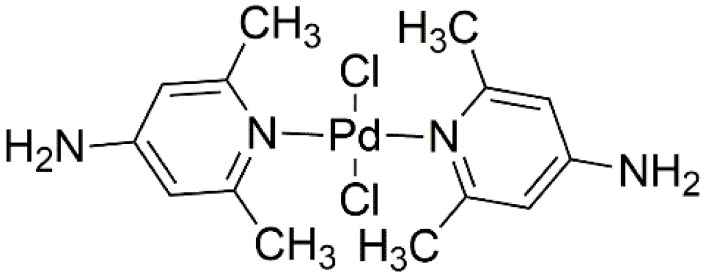
Structure of palladium derivative conjugated with 2,6-dimethyl-4-nitro-pyridine (dmnp).

**Table 1 ijms-22-08271-t001:** Recent and current clinical trials of platinum-based (mainly cisplatin, carboplatin, oxaliplatin) therapy—studies initiated in 2019–2021. This table was compiled from the information available at https://www.clinicaltrials.gov/ (accessed on 25 May 2021).

NCT Identification Number	Study Title	Clinical Trial Status	Type of Cancer	Study Design	References
NCT04300959	Anlotinib in combination with PD1 with gemcitabine plus(+)cisplatin for unresectable or metastatic biliary tract cancer	Phase 2 recruiting ^b^	Biliary tract cancer	Randomized, open-label, parallel assignment	[92]
NCT03912415	Efficacy and safety of BCD-100 (anti-PD-1) in combination with platinum-based chemotherapy with and without bevacizumab as first-line treatment of subjects with advanced cervical cancer (FERMATA)	Phase 3 recruiting ^a^	Cervical cancer	Randomized, double-blind (participant, investigator), placebo-controlled, parallel assignment	[93]
NCT04238988	Carboplatin-paclitaxel-pembrolizumab in neoadjuvant treatment of locally advanced cervical cancer (MITO CERV 3)	Phase 2 not yet recruiting ^b^	Locally advanced cervical cancer	Open-label, single-group assignment	[94]
NCT04428333	Study of GSK3359609 with pembrolizumab and 5-fluorouracil (5-FU)-platinum chemotherapy in participants with recurrent or metastatic head and neck squamous cell carcinoma (INDUCE-4)	Phase 2/3 active, not recruiting ^b^	Head and neck cancer	Randomized, double-blind (participant, investigator), placebo-controlled, parallel assignment	[95]
NCT04459715	A study of Debio 1143 in combination with platinum-based chemotherapy and standard fractionation intensity-modulated radiotherapy in participants with locally advanced squamous cell carcinoma of the head and neck, suitable for definitive chemoradiotherapy	Phase 3 recruiting ^b^	Squamous cell carcinoma of the head and neck	Randomized, quadruple-blind (participant, care provider, investigator, outcomes assessor), placebo-controlled, parallel assignment	[96]
NCT04517526	Efficacy and safety of platinum-based chemotherapy + bevacizumab + durvalumab, and salvage SBRT for IV non-small cell lung cancer patients with EGFR mutations after failure of first line osimertinib: a multicenter, prospective, Phase II clinical study	Phase 2 not yet recruiting ^b^	Lung cancer stage IV, EGFR-mutant, TKI, PD-L1, SBRT	Open-label, single-group assignment	[97]
NCT03467360	Inhibition of carbonic anhydrase in combination with platinum and etoposide-based radiochemotherapy in patients with localized small cell lung cancer (ICAR)	Phase 1 recruiting ^a^	Small cell lung cancer (SCLC)	Open-label, single-group assignment	[98]
NCT04774380	Study of durvalumab in combination with platinum and etoposide for the first line treatment of patients with extensive-stage small cell lung cancer (LUMINANCE)	Phase 3 not yet recruiting ^c^	Extensive-stage small cell lung cancer (SCLC)	Open-label, single-group assignment	[99]
NCT04660097	Anlotinib plus durvalumab-platinum-etoposide in first-line treatment extensive small-cell lung cancer	Phase 2 not yet recruiting ^c^	Extensive-stage small cell lung cancer (SCLC)	Open-label, single-group assignment	[100]
NCT04144608	Toripalimab combined with double platinum based chemotherapy for potentially resectable non-driver gene mutation non-small cell lung cancer	Phase 2 recruiting ^a^	Advanced non-small cell lung cancer (NSCLC)	Open-label, single-group assignment	[101]
NCT04612751	Datopotamab deruxtecan (Dato-DXd) in combination with durvalumab with or without platinum chemotherapy in subjects with advanced or metastatic non-small cell lung cancer (TROPION-Lung04)	Phase 1 recruiting ^b^	Advanced and metastatic non-small cell lung cancer (NSCLC)	Open-label, sequential assignment	[102]
NCT04324151	Pembrolizumab combined with double platinum based chemotherapy for potentially resectable NSCLC	Recruiting ^b^	Non-small cell lung cancer (NSCLC)	Retrospective observational cohort study	[103]
NCT04262869	Platinum-based chemotherapy and durvalumab for the treatment of stage IIIB or IV non-small cell lung cancer	Phase 2 recruiting ^b^	Non-small cell lung cancer (NSCLC)	Non-randomized, open-label, parallel assignment	[104]
NCT04832854	A study evaluating the safety and efficacy of neoadjuvant and adjuvant tiragolumab plus atezolizumab, with or without platinum-based chemotherapy, in participants with previously untreated locally advanced resectable stage II, IIIA, or select IIIB non-small cell lung cancer	Phase 2 recruiting ^c^	Non-small cell lung cancer (NSCLC)	Non-randomized, open-label, parallel assignment	[105]
NCT04351555	A study of osimertinib with or without chemotherapy versus chemotherapy alone as neoadjuvant therapy for patients with EGFRm positive resectable non-small cell lung cancer (NeoADAURA)	Phase 3 recruiting ^b^	Non-small cell lung cancer (NSCLC)	Randomized, double-blind (participant, investigator), placebo-controlled, parallel assignment	[106]
NCT03904108	Platinum-based chemotherapy plus ramucirumab in patients with advanced NSCLC who have progressed on first line anti-PD-1 immunotherapy	Phase 3 recruiting ^a^	Non-small cell lung cancer (NSCLC)	Open-label, single-group assignment	[107]
NCT04015778	A two-arm (Phase 2) exploratory study of nivolumab monotherapy or in combination with nab-paclitaxel and carboplatin in early stage NSCLC in China	Phase 2 recruiting ^a^	Non-small cell lung cancer (NSCLC)	Randomized, open-label, single-group assignment	[108]
NCT04586465	Dynamic PET/CT evaluated the response of neoadjuvant anti-PD1 combination with chemotherapy for Ⅱa–Ⅲb NSCLC (DYNAPET)	Phase 2 not yet recruiting ^b^	Non-small cell lung cancer (NSCLC), Stage IIA and IIIB	Open-label, single-group assignment	[109]
NCT04765709	Durvalumab and chemotherapy induction followed by durvalumab and radiotherapy in large volume stage III NSCLC (BRIDGE)	Phase 2 not yet recruiting ^d^	Non-Small Cell Lung Cancer (NSCLC), Stage III	Open-label, single-group assignment	[110]
NCT04158440	Study of toripalimab or placebo plus chemotherapy as treatment in early stage NSCLC	Phase 3 recruiting ^a^	Non-small cell lung cancer (NSCLC), Stage IIIA	Randomized, double-blind (participant, investigator), placebo-controlled, sequential assignment	[111]
NCT04676386	Biomarker analysis in high PD-L1 expressing NSCLC patients treated with PD-1/PD-L1 based therapy with or without the addition of platinum based chemotherapy (BEACON-LUNG)	Recruiting ^c^	Non-small cell lung cancer (NSCLC), Stage IIIC and IV	Prospective observational cohort study	[112]
NCT03912389	Efficacy and safety of BCD-100 (Anti-PD-1) in combination with platinum-based chemotherapy as first line treatment in patients with advanced non-squamous NSCLC (DOMAJOR)	Phase 3 recruiting ^a^	Non-squamous non-small cell lung cancer (NSCLC)	Randomized, double-blind (participant, investigator), placebo-controlled, parallel assignment	[113]
NCT04875611	Nivolumab in nasopharyngeal cancer with progression during or after platinum-based treatment (NIVONASO-21)	Phase 2 not yet recruiting ^d^(new)	Nasopharyngeal cancer	Open-label, single-group assignment	[114]
NCT03980925	Platinum-doublet chemotherapy and nivolumab for the treatment of subjects with neuroendocrine neoplasms (NENs) of the gastroenteropancreatic (GEP) tract or of unknown (UK) origin	Phase 2 recruiting ^a^	Neuroendocrine and Gastroenteropancreatic neuroendocrine cancer	Open-label, single-group assignment	[115]
NCT04851834	NTX-301 monotherapy in advanced solid tumors and in combination with platinum-based chemotherapy in advanced ovarian & bladder cancer and in combination with temozolomide in high-grade glioma	Phase 1/2 not yet recruiting ^c^	Advanced solid tumor; Platinum-resistant ovarian and bladder cancer; high-grade glioma	Non-randomized, open-label, sequential assignment	[116]
NCT04814875	A study to evaluate the combination of ATX-101 and platinum-based chemotherapy	Phase 1/2 not yet recruiting ^c^	Ovarian and Fallopian tube cancer; primary peritoneal carcinoma	Non-randomized, open-label, parallel assignment	[117]
NCT04274426	Mirvetuximab soravtansine (IMGN853), in folate receptor alpha (FRα) high recurrent ovarian cancer (MIROVA)	Phase 2 not yet recruiting ^d^	Recurrent epithelial ovarian, fallopian, or peritoneal carcinoma	Randomized, open-label, parallel assignment	[118]

^a^ the study started in 2019, ^b^ the study started in 2020, ^c^ the study started in 2021, ^d^ the study has yet to begin.

**Table 2 ijms-22-08271-t002:** Recent and current clinical trials on palladium-based therapy. This table was compiled from the information available at https://www.clinicaltrials.gov/ (accessed on 25 May 2021).

NCT Identification Number	Study Title	Clinical Trial Status	Type of Cancer	Study Design	References
NCT04480645	CivaDerm(TM) surface therapy pilot study	Early Phase 1 not yet recruiting ^a^	Basal and squamous cell carcinoma	Open-label, single-group assignment	[122]
NCT01106521	A registry study of permanent breast seed implant	Not applicable, recruiting	Breast cancer	Open-label, single-group assignment	[123]
NCT03290534	Feasibility study to treat lung cancer with the permanently implantable LDR CivaSheet^®^	Phase 2 recruiting ^b^	Lung cancer	Open-label, single-group assignment	[124]
NCT03109041	Initial feasibility study to treat resectable pancreatic cancer with a planar LDR source	Phase 1 recruiting ^b^	Pancreatic cancer	Open-label, single-group assignment	[125]
NCT00450411	Ultrasound-guided implant radiation therapy in treating patients with locally recurrent prostate cancer previously treated with external-beam radiation therapy	Phase 2 active, not recruiting ^+^	Prostate cancer	Open-label, single-group assignment	[126]
NCT00247312	Pd-103 dose de-escalation for early stage prostate cancer: a prospective randomized trial	Phase 3 completed ^a^	Prostate cancer	Randomized, open-label, parallel assignment	[127]
NCT02516709	Linear source registry for prostate cancer (CaRePC)	recruiting ^b^	Prostate cancer	Prospective observational cohort study	[128]
NCT00032006	Hormone therapy followed by internal radiation therapy in treating patients with locally recurrent prostate cancer	Phase 2 completed ^a^	Prostate cancer	Open-label, single-group assignment	[129]
NCT00023686	Surgery versus internal radiation in treating patients with stage II prostate cancer	Phase 3 completed ^a^	Prostate cancer	Randomized, open-label, parallel assignment	[130]
NCT00063882	Interstitial brachytherapy with or without external-beam radiation therapy in treating patients with prostate cancer	Phase 3 active, not recruiting ^+^	Prostate cancer	Randomized, open-label, parallel assignment	[131]
NCT01098331	Implant radiation therapy or surgery in treating patients with prostate cancer	Not applicable, unknown *	Prostate cancer	Randomized, open-label	[132]
NCT00006359	Androgen suppression plus radiation therapy in treating patients with prostate cancer	Phase 2 completed ^a^	Prostate cancer	Non-randomized, open-label, single-group assignment	[133]
NCT04033081	Registry of sarcoma patients treated with permanently implantable LDR CivaSheet^®^	Phase 4 recruiting ^b^	Sarcoma	Open-label, single-group assignment	[134]

^a^ The study was terminated. ^b^ The study is ongoing. ^+^ Results of the study are available on ClinicalTrials.gov. * No verification of status >2 years despite completion date has passed.

## Data Availability

Not applicable.

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
