# Peer review of "Platinum and Palladium Complexes as Promising Sources for Antitumor Treatments"

_ijms, 2021, doi:10.3390/ijms22158271_

Round 1

Reviewer 1 Report

The present review describes the state of the art relative to platinum and palladium complexes as chemotherapeutics. I found many interesting and new structures of these metal-based compounds and the relative bibliography is definitely updated. However, there are many revisions to address:

Introduction: I suggest the authors to better and more deeply clarify the differences between platinum and palladium from a chemical and a biological related point of view.

Paraggraph 2: In figure 1 the structures of complexes are not right. The coordination of the amino functions with platinum ion is a Lewis acid/base interaction. So a NH2----> Pt  bond is involved.

Paragraph 3: it is not clear the rationale chosen by the authors to select and present some platinum compounds among the others. It should be better if they add a rationale to explain it. Moreover, every time an acronym was indicated for both cancer lines and for specific targets, the author should spell out the meaning of the acronyms.

In reference to the treatment of glioblastoma (lines 169-170) I suggest the authors to introduce these reference: Biomedicine and Pharmacotherapy, 2018, 108, pp. 111–118.  In the paragraph related to the development multinuclear platinum complexes, I suggest the authors to clarify that these platinum complexes are cationic in nature and the corresponding mechanism of action. This reference could be added: European Journal of Inorganic Chemistry, 2019, 2019(29), pp. 3389–3395.

Paragraph 3.2: Are there any other intracellular target than DNA reported for palladium complexes? What are the main differences in the mechanisms of action of palladium complexes respect to platinum analogues? I suggest the authors to better clarify these aspects.

Line 264: AJ-5 is a complex with two Pd centres: it should be moved in the second part pf the paragraph (from line 336 forward).

Line 286-287: add a figure relative to this complex please.

Line 365: what is the active form of the complex? Clarify this aspect.

Paragraph 4: line 422: the botanical species is missing.

Only one example of Pt/Pd combination is proposed to make a conclusion. Please add some other examples.

Paragraph 5: please carefully check the English. Lines 481-482 do not make any sense to me. Lines 485-486: check them.

Line 534: explain the role of hyaluronic acid in this approach.

Lines 542-543: I would add some figures (TEM or SEM analyses)

References about the nanostructures used to improve pharmacokinetics of platinum and palladium complexes should be updated.

In general, I suggest the authors to revise the English of the manuscript or to make it read by a native English-speaking colleague. In alternative the authors could use the English Editing Service offered by MDPI.

I can suggest the publication of this manuscript only after these major revisions.

Author Response

Dear Editor,

Thank you for your letter of July 14th 2021, suggesting revision and resubmission of our manuscript entitled “Platinum and Palladium complexes as promising source for antitumor treatment” for publication in International Journal of Molecular Sciences.

We found the comments and suggestions of Reviewers helpful and have made revision accordingly. We hope that revised manuscript will satisfy Editor and Reviewers. We thank you for consideration and await your decision.

Please find attached response to Reviewer #1 and new version of the manuscript (the changes made in the manuscript are marked in red).

Sincerely,

Robert Czarnomysy

Reviewer 2 Report

The publication presented for evaluation (“Platinum and Palladium complexes as promising source for antitumor treatment”) is interesting material on the antitumor activity of the platinum and palladium compounds obtained so far. In my opinion, it can be accepted for publication after a few minor corrections.

  • I believe the authors should move compound 19 to chapter 3.2.
  • in the text of the publication, the authors refer to figures once, writing Figure no. bold, other times not, it would have to be standardized

I also propose introducing the following corrections:

Line 30: It was first change to It was the first

Line 93: Figure 1, structure 3 and 4 - Figure 1, structures 3 and 4

Line 134: Figure 4, structure 13 and 14 - Figure 4, structures 13 and 14

Line 142-143: complex with 2,6-bis(2-pyridyl)-4[1H]-pyridone and (14) 2,2’-((2- ([2,2’:6’,2’’-terpyridin]-4’-yloxy)ethyl)azanediyl)bis-(ethan-1-ol) - : complex with 2,6-bis(2-pyridyl)-4[1H]-pyridone and (14) 2,2’-((2-142 ([2,2’;6’,2’’-terpyridin]-4’-yloxy)ethyl)azanediyl)bis-(ethan-1-ol) (in such names we use; and not:, 1H is written in italics)

Line 160: in compound schemes, typically substituents are mentioned next to the formula, not in the figure caption

Line173: new three platinum complexes with 2,6-di-tert-butyl-4-(1-phenyl-1H-benzimidazol-2-2yl) phenol (Figure 6, structure 15), N, N-dimethyl-4-(1-phenyl-1H-benzimidazol-2-yl) aniline (Figure 6, structure 16) and 4-(1H-benzimidazol-2-yyl)-N, N-dimethylaniline - new three platinum complexes with 2,6-di-tert-butyl-4-(1-phenyl-1H-benzimidazol-2-2yl) phenol (Figure 6, structure 15), N,N-dimethyl-4-(1-phenyl-1H-benzimidazol-2-yl) aniline (Figure 6, structure 16) and 4-(1H-benzimidazol-2-yyl)-N,N-dimethylaniline  (1H italics, N,N- without space)

Similarly in the name of the figure 6

Reference list is not justified

Author Response

Dear Editor,

Thank you for your letter of July 14th 2021, suggesting revision and resubmission of our manuscript entitled “Platinum and Palladium complexes as promising source for antitumor treatment” for publication in International Journal of Molecular Sciences.

We found the comments and suggestions of Reviewers helpful and have made revision accordingly. We hope that revised manuscript will satisfy Editor and Reviewers. We thank you for consideration and await your decision.

Please find attached response to Reviewer #2 and new version of the manuscript (the changes made in the manuscript are marked in red).

Sincerely,

Robert Czarnomysy

Round 2

Reviewer 1 Report

The review has been extensively revised and improved in its contents. I'm completely satisfied by this version of the manuscript and the authors provided an exhaustive reply. In the present form, I suggest its publication in IJMS.